# Intercomparison of PurpleAir Sensor Performance over Three Years Indoors and Outdoors at a Home: Bias, Precision, and Limit of Detection Using an Improved Algorithm for Calculating PM_2.5_

**DOI:** 10.3390/s22072755

**Published:** 2022-04-02

**Authors:** Lance Wallace

**Affiliations:** Independent Researcher, Santa Rosa, CA 95409, USA; lwallace73@gmail.com

**Keywords:** low-cost particle monitors, PurpleAir, PM2.5, ALT-CF3, CF1, precision, limit of detection, Plantower, PMS-5003 sensors

## Abstract

Low-cost particle sensors are now used worldwide to monitor outdoor air quality. However, they have only been in wide use for a few years. Are they reliable? Does their performance deteriorate over time? Are the algorithms for calculating PM_2.5_ concentrations provided by the sensor manufacturers accurate? We investigate these questions using continuous measurements of four PurpleAir monitors (8 sensors) under normal conditions inside and outside a home for 1.5–3 years. A recently developed algorithm (called ALT-CF3) is compared to the two existing algorithms (CF1 and CF_ATM) provided by the Plantower manufacturer of the PMS 5003 sensors used in PurpleAir PA-II monitors. ***Results*.** The Plantower CF1 algorithm lost 25–50% of all indoor data due in part to the practice of assigning zero to all concentrations below a threshold. None of these data were lost using the ALT-CF3 algorithm. Approximately 92% of all data showed precision better than 20% using the ALT-CF3 algorithm, but only approximately 45–75% of data achieved that level using the Plantower CF1 algorithm. The limits of detection (LODs) using the ALT-CF3 algorithm were mostly under 1 µg/m^3^, compared to approximately 3–10 µg/m^3^ using the Plantower CF1 algorithm. The percentage of observations exceeding the LOD was 53–92% for the ALT-CF3 algorithm, but only 16–44% for the Plantower CF1 algorithm. At the low indoor PM_2.5_ concentrations found in many homes, the Plantower algorithms appear poorly suited.

## 1. Introduction

In recent years, a revolution in developing small low-cost particle sensors has occurred. A variety of sensors have been evaluated in multiple laboratory studies [1,2,3,4,5,6]. Their behavior in outdoor sites has also been studied extensively—a selection of these studies is provided [7,8,9,10,11,12]). The US Environmental Protection Agency has provided guidance on their use in outdoor settings [13,14]. However, fewer studies have focused on their use indoors [15,16,17,18,19,20,21].

Indoor studies are particularly relevant, since most people spend most of their time indoors [22,23]. To date, most epidemiology studies have been able to use only outdoor measurements to estimate human exposure; indoor exposures may or may not be estimated, but there are seldom any measured data [24,25,26]. Some studies such as the Harvard 6-City Study or the EPA’s Particle TEAM study have been able to use personal and indoor monitors to provide an estimate of the total personal or indoor exposure to particles [27,28,29,30,31]. However, these studies have been limited to relatively short-term exposures because of the expense of acquiring, setting up, and taking down expensive research-grade instruments in homes.

Now, it has finally become possible to determine *long-term* exposure to particles, since the new sensors are normally quiet and inconspicuous and can operate continuously without the need to maintain, clean, or manually download the data they collect. Downloading is made particularly easy by the PurpleAir company, Draper, Utah, USA (https://www2.purpleair.com/, accessed on 30 March 2022), which maintains a web-accessible database of all data except that for users who have requested privacy.

The overriding concern in using low-cost particle sensors is their reliability and accuracy. Accuracy is a function of bias and precision. However, bias can be corrected; precision cannot.

The manufacturer of the PMS 5003 sensors used in PurpleAir PA-II monitors is the Plantower company (http://www.plantower.com/en/, accessed on 30 March 2022). Plantower employs two proprietary algorithms (CF1 and CF_ATM) to estimate PM_2.5_. For PM_2.5_ concentrations less than approximately 28 µg/m^3^, which constitute the vast majority of measured concentrations in many nations, the two algorithms give identical results [19]. Plantower provides no information about the composition or indeed the very existence of a calibration aerosol. Nor does the company provide any other data about the calculations involved in translating observed numbers of particles in six size categories into a mass concentration estimate. We have shown how both these algorithms are seriously flawed. We developed a transparent and reproducible algorithm (ALT-CF3) that was shown to outperform the Plantower algorithms with respect to bias, precision, limit of detection, and distribution fitting [19,32]. Our results suggest that the Plantower algorithms overestimate PM_2.5_ concentrations by approximately 40–50%, as has been found by other investigations [1,3,8,11,32]. However, this bias can be corrected. What is much more serious is the Plantower problem with precision.

We show in this paper that estimates of precision may be severely impacted by the flawed Plantower practice of assigning a value of zero to concentrations not reaching a certain predefined threshold. In particular, we show that indoor concentrations in a home with reasonably typical indoor activities such as cooking, cleaning, and other particle-producing activities fall below this threshold for such a large fraction of time that estimates of both precision and PM_2.5_ concentration are limited to a greatly reduced fraction of all observations. We also use our 18 month or 3 year data collection periods to investigate the question of how the performance may change over extended periods. We also compare the LODs of the Plantower and ALT-CF3 algorithms, and show the fraction of outdoor and indoor measurements falling below the LODs for each algorithm.

## 2. Materials and Methods

All data were collected inside or outside a home with two residents. The indoor PurpleAir monitors were placed on a desk, dresser, or cart approximately 1.0 m high (Appendix A). The outdoor monitor was hung approximately 2 m high from a bracket approximately 2 m high and 15 cm from the home (Appendix A). The home is a detached 1-story building of approximately 400 m^3^ volume located in the city of Santa Rosa, CA, USA. For the 18-month period from 10 January 2019 to 18 June 2020, two indoor PurpleAir monitors collected data every 80 s (later, every 2 min). For the next 18-month period (18 June 2020–14 January 2022), two additional PurpleAir monitors collected both indoor and outdoor data.

Heating is supplied by a natural gas furnace, and the temperature is normally set to 72 °F (22 °C). A gas stove and electric oven are used for cooking on approximately 5 or 6 days a week. A central air conditioner is used at times in the summer. The central fan is normally on and the main filter is equipped with electrically charged wires to attract particles. The air exchange rate of the home was measured on multiple occasions and was generally in the range from 0.2 to 0.3 air changes per hour (80–120 m^3^/h). Two blower door tests confirmed that the home was well constructed with a low baseline air change rate. Windows are normally closed except in summer, when the air conditioner is not being used. Some experiments were performed in a closed room involving single puffs from a vaping pen containing marijuana liquid. These experiments resulted in increased PM_2.5_ levels for less than 4% of the time. Additional experiments were performed using three co-located research-grade SidePak AM510 optical particle monitors (TSI Inc., Shoreview, MN, USA (https://tsi.com/home/, accessed on 30 March 2022)). The SidePak monitors were equipped with a cleaned and greased PM_2.5_ impactor. Flow rates were set at 1.7 Lpm using a TSI 4140 inline flow meter. Approximately 18 typical indoor sources were examined. Cooking sources such as broccoli, chicken, and hamburger were heated using miniature pans (cast iron, stainless steel and coated pans) on a laboratory hotplate (Cimarec Model SP-131015, ThermoFisherScientific, Waltham, MA, USA (https://www.thermofisher.com/us/en/home.html, accessed on 30 March 2022)) to temperatures sufficient to create particle numbers above background. In some cases, extended time of heating was used to provide blackened shrimp or burnt toast (Appendix A).

### The ALT-CF3 Algorithm

The central method of estimating PM_2.5_ concentrations from PurpleAir monitors was explored in two papers [19,32]. The method is well known and has been used for many years by persons working with optical particle monitors or companies manufacturing optical monitors such as Climet (http://www.climet.com/, accessed on 30 March 2022), MetOne (https://metone.com/, accessed on 30 March 2022), and TSI (https://tsi.com/, accessed on 30 March 2022). Optical particle monitors often provide particle counts for several size categories, 0.3–0.5 µm, 0.5–1 µm, and 1–2.5 µm. There are 3 additional larger categories but they do not contribute to PM_2.5_. Given the number N of particles in a size category, one can estimate the total volume of the particles by choosing a diameter *d* within the range of the size category, and calculating the total volume V associated with that diameter d using the equation *V* = *Nπ d*^3^/6. The diameter *d* is typically either the arithmetic or geometric mean of the boundaries of the size category. For example, the diameter of the 0.3–0.5 µm category is either the arithmetic mean (0.4 µm) or the geometric mean (0.37 µm). Once the total volume of all of the desired size categories is determined, multiplying by the density of the aerosol mixture provides the mass (e.g., PM_1_ if only the two smallest categories are employed, or PM_2.5_ if the three smallest size categories are used). In the case of the PurpleAir monitors, the chosen diameter was the geometric mean and the density chosen for the particles was that of water (1 g/cm^3^) [19,32]. The density of typical PM_2.5_ particles depends on local conditions, but is often taken to be in the range of 1–2 g/cm^3^. The actual choice of the density is unimportant, because the ultimate calibration of the monitors will depend on comparisons with reference monitors, and that comparison will give the calibration factor (CF) required to bring the PurpleAir estimate into agreement with the reference monitor.

The first application of this method to PurpleAir monitors was provided in 2020 [19]. In that paper, PM_2.5_ aerosols produced by vaping marijuana liquid were measured using two PurpleAir PA-II monitors (four Plantower sensors) co-located with three SidePak (TSI) and two Piezobalance (Kanomax, Osaka, Japan, https://kanomax.biz/asia/, accessed on 30 March 2022) research-grade monitors. The SidePaks and Piezobalances were calibrated by comparison with a gravimetric system employing a pump, filter, and microbalance at Stanford University [19]. The ultimate calibration factor for the PurpleAir monitors was 3.0 (SE = 0.015), based on 47 experiments carried out over one year (Appendix A). This CF applies to the indoor aerosol mixture encountered during the experiments, which took place in a room of a detached house and included PM_2.5_ from typical household activities together with the PM_2.5_ produced by vaping marijuana liquid.

The next application of the calibration method took place in 2021 [32]. In this study, 33 Purple Air sites within 500 m of 27 EPA regulatory (Federal Reference Method (FRM) or Federal Equivalent Method (FEM)) monitors were compared over extended periods of time (177,000 hourly average measurements). The same method of calculating PM_2.5_ as in [19] was adopted. Since this was an ALTernative to the Plantower algorithms, it was given the name ALT-CF3. Four different analytical approaches resulted in an estimated CF of 3.05 (SE 0.05). The ALT-CF3 method can be applied to any and all operating PurpleAir monitors accessible on either the PurpleAir API website (https://api.purpleair.com/, accessed on 30 March 2022) or the main PurpleAir map website (https://www2.purpleair.com/, accessed on 30 March 2022). The ALT-CF3 algorithm is presently the only one of several alternative methods available on those sites that does not depend on the Plantower algorithms.

It is important to note that the performance of the Plantower sensors is not necessarily due to the accuracy of the particle counts. In fact, there is much evidence that the counts are NOT accurate (e.g., [2]). There is even evidence from the ALT-CF3 results that the particle numbers are probably underestimated. That is because a multiplier of 3 must be applied to a calculation using an assumed density of 1 g/cm^3^, whereas PM_2.5_ is expected to have a density on the order of 1–1.5 g/cm^3^. If so, the multiplier of 3 used to match results to the reference monitors must indicate a underestimate of particle numbers by approximately a factor of 2 or 3. This is an example of the way that a bias, if reasonably constant, can be corrected to provide good accuracy—but only if the precision is good, as it is for the ALT-CF3 algorithm.

In the following sections, to save space (particularly in tables), we will often use “CF3” as a synonym for “ALT-CF3”.

PurpleAir PA-II monitors include two independent Plantower PMS 5003 sensors, which we identify as *a* and *b*. We define precision for each monitor as the absolute difference between the PM_2.5_ readings divided by their sum: abs(*a* − *b*)/(*a* + *b*). The monitors employed in this study are identified by the numbers 1–4. For example, the two sensors in monitor 1 are identified as 1*a* and 1*b*.

## 3. Results

Monitors 1 and 2 collected data for 3 years. Monitor 1 was indoors the entire time; monitor 2 was indoors for 2 years and outdoors for 1 year. A total of 829,907 observations were made; some were 80 s averages, most were 2 min averages. Only the ALT-CF3 values with a precision better than 20% (under 0.2) were accepted. This left 92% of the data (763,102 observations).

Monitors 3 and 4 collected data over the last 18 months of this study (18 June 2020 to 14 January 2022). Monitor 3 was outdoors for all but one month; monitor 4 was indoors the entire time. There were 406,310 observations in total, and 370,906 observations remained after limiting the data to those with a precision under 0.2. There were 353,256 matched pairs of indoor–outdoor PM_2.5_ measurements. They appeared to be close to log-normally distributed, spanning approximately 5 orders of magnitude from 0.01 to 450 μg/m^3^ (Figure 1).

A log-normal distribution would form a straight line on the graph. The departure from log-normal behavior at the higher concentrations may be due to the wildfires affecting this northern California site in 2021. There appears to be a fairly constant fraction relating indoor to outdoor concentrations.

The same observations were plotted using the Plantower CF1 algorithm (Figure 2). More than 50,000 indoor measurements (17% of the total) and 14,000 outdoor measurements fell below the cutoff value of 0.01 at which the Plantower algorithm assigns a value of zero. This accounts for the missing data between approximately −5 and −2 normal probability standard deviations.

It should be noted that although the Plantower CF1 cutoff is at 0.01 µg/m^3^, the ALT-CF3 data for these same observations shows 20–40 times the value of 0.01 at a Z-score of −2. The ALT-CF3 algorithm never returns a value of zero, since particles are always present in the smallest size category (0.3–0.5 µm). Because of the lack of information for the Plantower CF1 algorithm, it is not known why so many observations are assigned a value of zero.

It can also be seen that the distribution of concentrations using the Plantower CF1 algorithm is affected by the many zeros and cannot be fitted with a log-normal curve.

The PM2.5 measurements for all monitors and both time periods are supplied in Appendix A. Mean indoor PM_2.5_ values ranged from 3.6 to 5.7 µg/m^3^. These values are quite comparable to those found in a previous study including 91 PurpleAir indoor monitors averaged over times from 796 to 13,564 h. The observed means ranged from a median of 3.4 µg/m^3^ to a 75th percentile value of 5.5 µg/m^3^ [15].

### 3.1. Relative Bias

In this section, we compare the bias relative to the mean of all eight sensors, first for the ALT-CF3 algorithm and then for the Plantower CF1 algorithm. We look at both indoor and outdoor measurements. The bias relative to the mean of ALT-CF3 estimates of PM_2.5_ (µg/m^3^) for sensors *a* and *b* was calculated for the final 18 month period when all 4 monitors (8 sensors) were operating. The mean (SE) overall bias was 3.0% (0.7%), ranging from 0.5% to 5% (Appendix A).

The same calculations of bias for the Plantower CF1 algorithm resulted in the loss of more than 100,000 observations for the indoor monitors 1 and 4 due to the large number of reported zeros for the PM_2.5_ observations. Both indoor and outdoor readings from monitor 2 lost approximately 50% of all observations. Monitor 3 lost more than 75,000 observations (Appendix A).

There was also an effect of increased mean values even beyond the 40% overestimate often noted for the Plantower CF1 algorithm. The mean indoor values were approximately 100% higher than for the ALT-CF3 algorithm, and the mean outdoor values were more than 200% higher. The mean (SE) overall bias was 4.0% (1.5%) ranging from 0.6% to 8.2%. These bias calculations are only slightly worse for the Plantower CF1 algorithm, because they are “helped” by not having to consider the thousands of deleted observations which would otherwise lead to different (probably higher) estimates of bias.

The absolute bias of these four monitors with respect to regulatory FEM or FRM monitors could not be determined over the three years of this study, since no regulatory monitor was nearby. However, a previous study compared 33 PurpleAir outdoor monitors to 27 nearby regulatory monitors [32]. The bias (ratio of PurpleAir monitor PM_2.5_ using the ALT-CF3 calibration to regulatory monitor PM_2.5_) had a median value of 0.96 (IQR 0.77 to 1.21).

#### Comparison with FEM Bias

The US EPA operates a program determining the bias of selected Federal Equivalent Method (FEM) PM_2.5_ results by setting up a side-by-side gravimetric monitor employing the Federal Reference Method (FRM) (https://www.epa.gov/outdoor-air-quality-data/pm25-continuous-monitor-comparability-assessments, accessed on 30 March 2022). The bias of the continuous FEM monitor with respect to the collocated FRM monitor is calculated over a 3 year period. A sample of 61 reports from the state of CA showed a mean absolute bias of 20.5% (SE 3.2%) for the FEM monitor compared to the FRM monitor. 41 of the calculated biases were positive, with a mean (SE) of +27.3% (4.5%). The median bias was 1.21 (IQR 1.04 to 1.25). These results suggest that the range of the absolute bias of 33 PurpleAir monitors employed in [32] using the ALT-CF3 algorithm appears comparable to the absolute bias of the 61 FEM monitors compared to FRM monitors in the EPA comparability assessments.

### 3.2. Precision

For the entire 3 year period, median precision for monitors 1 and 2 was between 4.6% and 5.7% using the ALT-CF3 algorithm (Table 1). The Plantower CF1 median precision for the same data ranged between 8.4% and 20.5% with, however, more than 100,000 fewer measurements for monitor 1 and 50,000 fewer for monitor 2 due to the excessive number of zeros reported by the CF1 algorithm.

Applying the precision cutoff of 0.2 for the Plantower CF1 measurements resulted in an additional loss of 25% of the remaining data for monitor 1 and 45–50% of the remaining data for monitor 2. Overall, the loss of data using the Plantower CF1 algorithm amounted to 277,488 (36%) measurements for monitor 1 and 387,991 (52%) of all measurements for monitor 2 (Figure 3).

The loss of observations over a 3 year period for the Plantower CF1 algorithm compared to the ALT-CF3 algorithm was 36% for monitor 1 and 52% for monitor 2.

For the 18 month second period employing monitors 3 and 4, the loss of data due to employing the CF1 algorithm was similar for the indoor data at 33%, but improved for the outdoor data (20%) (Appendix A).

### 3.3. PM_2.5_ Concentrations of Zero

From the 3 year and 18 month studies, the Plantower CF1 algorithm reported PM_2.5_ concentrations of zero for approximately 60,000 to 160,000 indoor measurements (12% to 23% of the total) and 10,000 to 35,000 outdoor measurements (4–14% of the total) (Table 2). The ALT-CF3 algorithm, by contrast, reports *no* concentrations of zero for the same set of observations. This is because there are never occasions when the number of particles in the smallest size category (0.3–0.5 um) falls to zero.

The prevalence of zeros produced by the Plantower CF1 and CF_ATM in indoor data, ranging from 12 to 23% of the observations, is due entirely to the Plantower decision to define all values below a certain cutoff as zero. The cutoff appears to be 0.01 µg/m^3^ as defined by both the Plantower CF1 and CF_ATM algorithms. However, we found that only a minute percentage (15 of 50,000) of these same observations were below 0.01 µg/m^3^ as defined by the ALT-CF3 algorithm. Instead, the ALT-CF3 measurements of these same observations ranged up to 30 µg/m^3^, although most were below 1 µg/m^3^. We can find no apparent reason for this result, which is certainly not reflected by the particle numbers reported in the three size categories. This loss of data will be impossible to recover for all investigators using either of the Plantower CF1 and CF_ATM algorithms, no matter what further modifications they apply to the Plantower algorithms. Statisticians do not approve of replacing concentrations below the LOD with zero, the LOD itself, or half the LOD [33]. In this case, the cutoff is in fact far below the LOD, as discussed in Section 3.5.

### 3.4. Variation of Precision over Time

We regressed the measured precision against time of operation (3 years for monitors 1 and 2; 18 months for monitors 3 and 4), treating indoor and outdoor measurements separately. Results were split evenly, showing increased precision in three cases and decreased precision in the remaining three cases (Table 3).

### 3.5. Limit of Detection (LOD)

A definition for the LOD for the case of analyzing a physicochemical sample can be found in many publications (e.g., [34]). The definition envisions analyzing several samples expected to have concentrations near the LOD. If the results of analyzing the several samples shows that the mean is more than 3 times the standard deviation, then the mean value is considered to be near (somewhat above) the LOD.

For the case of continuous sampling, a different definition is needed. One approach was advanced in [18]. In this definition, the LOD occurs at the lowest mean value µ above which more than 95% of mean values exceed their standard deviations σ by more than a factor of 3 (*µ*/*σ* > 3). In practice, this requires identifying all cases with *µ*/*σ* < 3, sorting according to *µ* ascending, and then counting the number of cases with *µ*/*σ* < 3 in, say, blocks of 100 in ascending order. When a block of 100 is reached with fewer than 5*µ*/*σ* values < 3, a candidate for the LOD has been found within that block. However, further exploration at higher mean concentrations may show a new block of 100 with 5 or more cases of *µ*/*σ* < 3, at which point that new block of 100 now contains a new (higher) candidate for the LOD. The search ends when all the data have been explored, but in practice it ends much earlier, when there are increasingly great “distances” between blocks of 100 with 5 or more values of *µ*/*σ* < 3.

LODs were determined for all four monitors. For monitor 2, which spent time indoors and outdoors, the LOD was calculated separately for each location. The LODs calculated for the ALT-CF3 algorithm ranged from 0.6 to 1.3 µg/m^3^, compared to from 2.9 to 9.9 µg/m^3^ for the Plantower CF1 algorithm (Table 4). Approximately 53–92% of the data exceeded the CF3 LODs, but only 16–44% of the data exceeded the Plantower CF1 LODs (Figure 4).

### 3.6. Comparison with Co-Located Research-Grade SidePak Monitors

Typical indoor particle sources were examined by the three PA-II PurpleAir monitors co-located with three SidePak AM510 monitors. These sources included cooking oils (butter, olive oil, and coconut oil); vegetables (asparagus, broccoli, and red pepper); meats (blackened shrimp, chicken, and hamburger); wooden kitchen matches; dust or SVOCs collected over more than a year on coated or stainless steel pans or on glass Petri dishes; and burnt toast. Outdoor aerosol was also studied; however, during the time of study, the outdoor air was very clean, so the CFs were not tested over a reasonable range of concentrations. Most of the cooking sources were placed on three cleaned miniature pans: cast iron, coated, and stainless steel. Before cleaning, the pans were heated using the hotplate to temperatures of 400–500 °C to drive off the collected dust. Temperatures were gradually increased for each source until particles began to be observed. The hotplate was then turned off without further increase, except in the cases when blackened foods were desired (e.g., blackened shrimp, broccoli, and burnt toast). The Plantower and ALT-CF3 algorithms were applied to the results, requiring all PurpleAir precision to be better than 20%. This requirement resulted in a loss of approximately 8% of the original 2300 2 min averages to 2158 valid observations for the ALT-CF3 algorithm and a much larger loss of almost 50% to 1209 valid observations for the Plantower CF1 algorithm. A particularly striking loss of data occurred for burning wooden kitchen matches—from 256 measurements for the ALT-CF3 algorithm down to 16 measurements for the Plantower CF1 algorithm, a loss of approximately 95% of all data. This loss was not due to the Plantower CF1 assignment of zero to measurements below the threshold, since there were only 15 such cases, the concentrations in general being quite high. The ratios for each of the two algorithms with the SidePak results for all 17 sources are shown in Figure 5. The CFs for the ALT-CF3 algorithm are typically near 0.4. This would be expected for an aerosol mixture with a density near 1 g/cm^3^, because the SidePak is calibrated with Arizona Road Dust having a density of 2.6 g/cm^3^ (1/2.6~0.4). Although the CF depends on other characteristics (refractive index, RH, and particle composition), density may in many instances be the controlling factor. The main exception to the general CF of 0.4 was for the ignition of paraffin wax, which resulted in a short-lived fire and produced an estimated CF of only 0.16. However, this is well explained by the fact that the SidePaks are able to handle occupation-level high concentrations (which went in this case to a maximum of 12 mg/m^3^) whereas the PurpleAir monitors have an upper limit of approximately 1 mg/m^3^. In keeping with the general tendency of the CF1 algorithm to overpredict PM_2.5_ concentrations, the CFs reported by the CF1 algorithm are higher (mean approximately 0.7) and more variable (0.6–0.8). The error bars are also larger for the CF1 algorithm.

### 3.7. Limitations

Only four monitors and only one location was tested in this study. Other long-term studies in other locations would be desirable.

## 4. Discussion

All major findings regarding the Plantower CF1 algorithm stand on their own and do not depend on the calibration factor of 3 used in the ALT-CF3 algorithm. These findings for the Plantower CF1 algorithm include (1) the assignment of zero to values smaller than a cutoff concentration; (2) the resulting loss of data, which increases at lower concentrations such as those encountered indoors; (3) a resulting decline in precision; (4) a high LOD in the 3–10 µg/m^3^ range, leading to major portions of datasets falling below the detection limit.

The generally good accuracy shown by the PurpleAir monitors is not an indication of the accuracy of the particle counts themselves. In fact, there is evidence that the particle counts are underestimated by a factor of approximately 2 [19]. However, because the precision of the sensors is good to excellent, this bias appears to be correctable sufficiently to give good results when compared to reference monitors.

Regarding the question of possible decline over time of sensor response, our 3 year period showed improved precision for three sensor/locations and declining precision for the other three sensor/locations, so presented little evidence for a general decline.

The comparisons with the research-grade SidePak indicated considerable uniformity and stability of the ALT-CF3 algorithm, with the PurpleAir/Sidepak ratio not far from 0.4 for 16 of the 17 sources. The 17th source (the fire due to paraffin wax ignition) resulted in exceeding the upper limit of approximately 1 mg/m^3^ for the PurpleAir monitor and therefore the observed ratio of 0.16 is not an exception to that observation.

The comparisons with the research-grade SidePak monitors do not in themselves have any bearing on the accuracy of the PurpleAir monitors, since that would require concurrent filter collection followed by gravimetric methods. This cannot reasonably be performed over a period of an hour or so since not enough weight would collect on the filters. However, several studies have measured SidePak CFs for different particle sources. Perhaps the most important source to be studied to determine calibration factors is outdoor air, even when indoor air is also being studied. This is because air of ambient origin is normally a substantial contributor to the total indoor aerosol mixture. Unfortunately, the study most focused on establishing calibration factors for the SidePak for outdoor air found the CFs in 18 cases over several months to vary from 0.31 to 1.05 [35]. The authors concluded that perhaps outdoor air is so changeable in composition, refractive index, density, and size distributions that no single CF will be sufficient to characterize the SidePak response to outdoor aerosol.

A slope of 3.38 was found for the SidePak based on comparisons with reference monitors for ambient air [36]. This would translate to a CF of 0.29 (1/3.38) if the intercept were close to zero, but in fact the intercept was 5.8 µg/m^3^, so the slope might change if forced through zero.

A study of 1400 buildings taken from the PurpleAir sensor network in two California areas during times of wildfires calculated infiltration ratios for days influenced or not influenced by fires [37]. The ratios were lower during wildfire days indicating that residents took action to protect themselves from the worsened outdoor air. The authors identified and removed periods when indoor sources were evident. They identified 16 regulatory monitoring sites within 5 km of their selected outdoor PurpleAir sites and found a calibration factor of 0.53 for the Plantower CF1 algorithm. This suggests that the CF1 algorithm overestimates PM_2.5_ concentrations by nearly a factor of 2, somewhat more than other measures of approximately 40–50% overestimates. However, it is in close agreement with the CF of 0.48 found in another study of wildfire smoke [38].

A laboratory study of responses to 24 common indoor sources for 7 selected low-cost monitors including PurpleAir PA-II monitors compared to two research monitors was performed [4]. Although the authors did not attempt to determine calibration factors for the monitors, they did estimate the densities of the aerosol mixtures, which would affect the calibration factors. The densities varied widely, which would indicate that the CFs would also vary widely according to the dominant indoor source at any time. It may be worth quoting their final paragraph:

“The evaluated versions of the AirBeam, AirVisual, Foobot, and Purple Air II monitors were of sufficient accuracy and reliability in detecting large sources that they appear suitable for measurement-based control to reduce exposures to PM_2.5_ mass in homes. The logical next steps in evaluating these monitors are to study their performance in occupied homes and to quantify their performance after months of deployment.”

This study is a first step toward carrying out their recommendation.

## 5. Conclusions

At typical levels of both indoor and outdoor PM_2.5_, the Plantower CF1 and CF_ATM algorithms lose an unacceptably large fraction of observations due to the choice of setting a threshold below which observations are assigned a value of zero. This practice is not supported by statistical theory. It causes the loss of substantial amounts of data (12–23% in our three-year study). It also increases the already high overestimates of PM_2.5_ due to deleting lower concentrations. No models employing either of the two Plantower algorithms can restore these values. The ALT-CF3 algorithm loses no data, since it depends on particle counts that never go to zero. The ALT-CF3 algorithm also has improved precision and limits of detection.

Mean precision using the CF3 algorithm is 5–6%, compared to 8–20% for the Plantower CF1 algorithm. In addition, there is a very great loss of data due in part to the choice to substitute zero for all values below a certain threshold. If an upper limit is set for the precision, the mean precision for both algorithms is improved, but again the loss of data was in the hundreds of thousands of observations for the Plantower CF1 algorithm.

With respect to the LOD, there was a very large difference between the two algorithms, with the LODs for the Plantower CF1 algorithm so high (3–10 µg/m^3^) that more than half of all data (56–84%) was below the LOD. By contrast, the ALT-CF3 algorithm produced LODs generally below 1 µg/m^3^, resulting in more than half of all data (53–92%) *above* the LOD.

Since the PM_2.5_ measurements using the ALT-CF3 algorithm are readily available on the PurpleAir map site and the API site, there is no bar to choosing the ALT-CF3 algorithm for future studies employing the PurpleAir data from any time period.

## Figures and Tables

**Figure 1 sensors-22-02755-f001:**
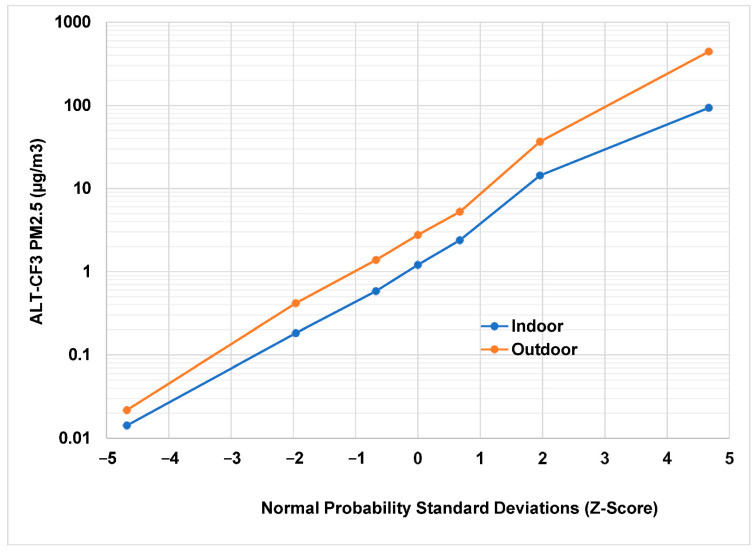
Indoor and outdoor PM_2.5_ concentrations (N = 353,256) over 18 months using the ALT-CF3 algorithm. The three middle points centered on the median at 0 provide the interquartile range (25th and 75th percentiles).

**Figure 2 sensors-22-02755-f002:**
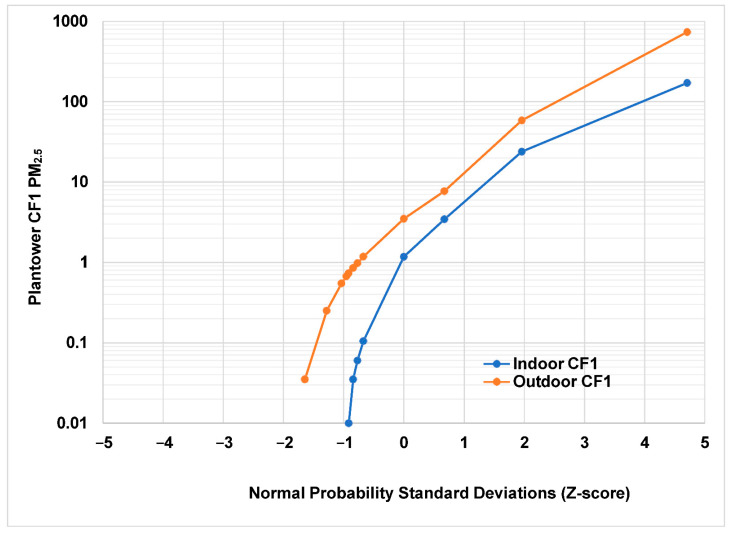
Same observations as in Figure 1 using the Plantower CF1 algorithm. Many measurements have been assigned a value of zero and cannot be shown on the logarithmic graph.

**Figure 3 sensors-22-02755-f003:**
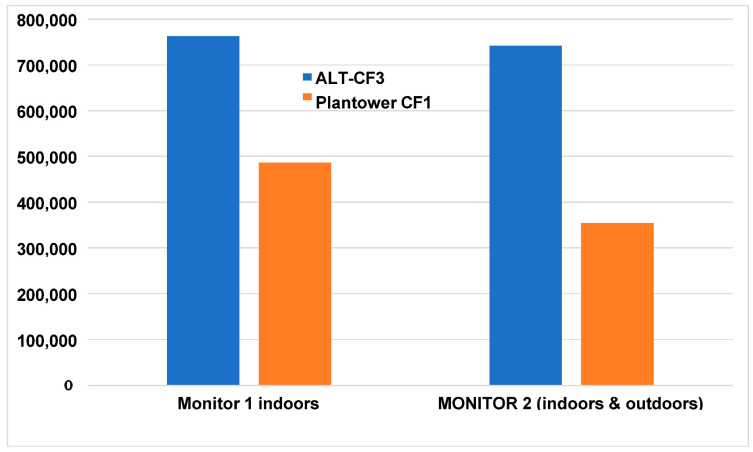
Total observations remaining after applying an upper precision limit of 0.2 (20%).

**Figure 4 sensors-22-02755-f004:**
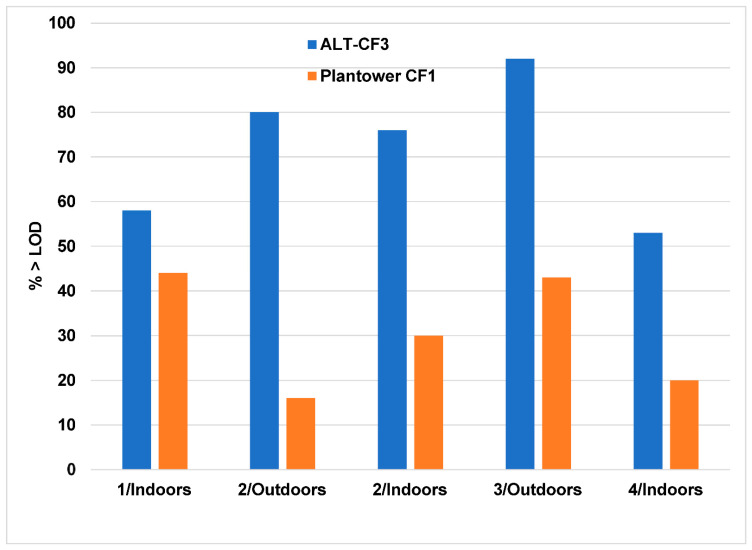
Percent of observations exceeding the LOD compared for the ALT-CF3 and Plantower CF1 algorithms. Monitor/Location shown on x-axis.

**Figure 5 sensors-22-02755-f005:**
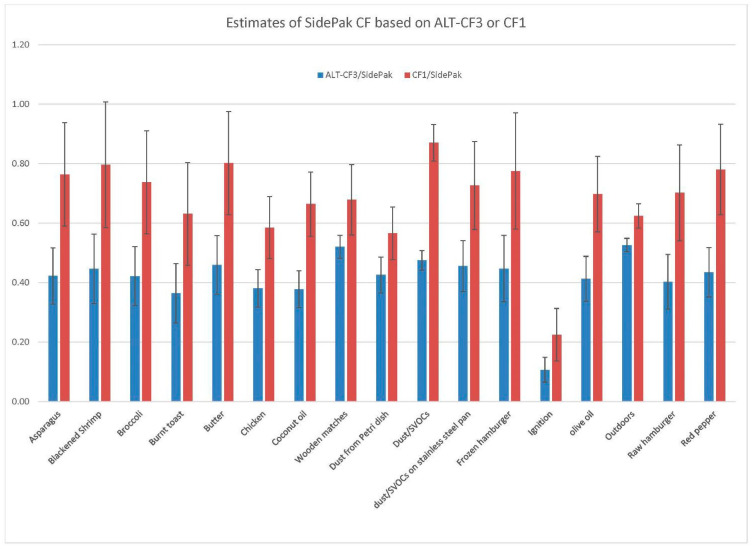
Ratios of the ALT-CF3 and Plantower CF1 PM_2.5_ estimates with the co-located SidePak estimates for 17 sources. Error bars are propagated standard errors.

**Table 1 sensors-22-02755-t001:** Precision compared using ALT-CF3 and Plantower CF1 algorithms. Time period: 1 October 2019 to 14 January 2022.

	Valid N	Mean	Std. Err.	Lower Quartile	Median	Upper Quartile	90th %Tile	Max
** *ALT-CF3 algorithm (using precision cutoff of 0.2)* **								
Monitor 1 indoors	763,102	0.064	0.000055	0.025	0.053	0.094	0.14	0.2
Monitor 2 indoors	499,296	0.067	0.000068	0.027	0.057	0.097	0.14	0.2
Monitor 2 outdoors	242,663	0.058	0.000093	0.021	0.046	0.084	0.13	0.2
** *Plantower CF1 algorithm (using ALT-CF3 cutoff of 0.2)* **								
Monitor 1 indoors	647,757	0.192	0.000334	0.034	0.084	0.20	0.57	1
Monitor 2 indoors	448,867	0.337	0.000495	0.072	0.205	0.51	1	1
Monitor 2 outdoors	234,814	0.293	0.000631	0.065	0.172	0.41	0.92	1
** *Plantower CF1 algorithm (using precision cutoff of 0.2)* **								
Monitor 1 indoors	486,614	0.067	0.000074	0.025	0.055	0.10	0.15	0.2
Monitor 2 indoors	224,877	0.081	0.000118	0.033	0.071	0.13	0.17	0.2
Monitor 2 outdoors	129,081	0.082	0.000157	0.033	0.073	0.13	0.17	0.2

**Table 2 sensors-22-02755-t002:** Number of PM_2.5_ concentrations reported as zero by the Plantower CF1 algorithm for monitors 1 and 2 over a 3 year period and monitors 3 and 4 over an 18 month period.

Sensor	Location	N Obs.	N Zeros	Fraction = 0
1a	Indoors	815,558	165,732	0.20
1b	Indoors	817,696	164,399	0.20
2a	Indoors	530,781	63,867	0.12
2b	Indoors	558,322	130,263	0.23
4a	Indoors	406,059	61,444	0.15
4b	Indoors	406,068	69,435	0.17
2a	Outdoors	252,532	10,324	0.04
2b	Outdoors	253,439	35,374	0.14
3a	Outdoors	363,786	23,516	0.06
3b	Outdoors	363,783	18,757	0.05

**Table 3 sensors-22-02755-t003:** Variation of precision over time for monitors 1 and 2 (3 years) and 3 and 4 (18 months).

	3 Year Period (10 January 2019 to 14 January 2022)			18 Month Period (18 June 2020 to 14 January 2022)		
Monitor	1 IN	2 IN	2 OUT	3 OUT	3 IN	4 IN
Location	Indoors	Indoors	Outdoors	Outdoors	Indoors	Indoors
N	763,102	499,296	242,663	356,484	42,204	370,906
Intercept	−0.28	−0.33	0.61	−0.27	1.6	0.1
SE (Int.)	0.007	0.010	0.040	0.019	0.039	0.022
Slope	7.8 × 10^−^^6^	9.0 × 10^−^^6^	−1.2 × 10^−^^5^	7.4 × 10^−^^6^	−3.4 × 10^−^^5^	−8.7 × 10^−^^7^
SE (slope)	1.7 × 10^−^^7^	2.3 × 10^−^^7^	9.1 × 10^−^^7^	4.3 × 10^−^^7^	8.8 × 10^−^^7^	4.8 × 10−^7^
R^2^ (adj.)	0.0028	0.00319	0.00076	0.00082	0.034	0.00006
SE of estimate	0.048	0.048	0.046	0.042	0.032	0.050
F-value	2181	1599	186	296	1500	3.2
z	47	40	−14	17	−39	−2
*p*-value	0	0	0	0	0	0.072
starting precision	0.060	0.062	0.083	0.054	0.058	0.068
ending precision	0.068	0.072	0.070	0.058	0.038	0.067
Relative annual increase (%)	4.8	5.3	−5.3	5.3	−22.6	−0.49

**Table 4 sensors-22-02755-t004:** PM_2.5_ LODs (µg/m^3^) calculated for the ALT-CF3 and Plantower CF1 algorithms. Number and percent of observations greater than the LOD.

Sensor	Location	Valid N	CF3 LOD	# Obs with CF3 > LOD	% Obs with CF3 > LOD	CF1 LOD	# Obs with CF1 > LOD	% Obs with CF1 > LOD
1	Indoors	406,108	0.99	233,900	58	2.9	177,908	44
2	Outdoors	253,454	0.92	203,384	80	9.9	39,487	16
2	Indoors	146,229	0.72	110,674	76	3.2	44,289	30
3	Outdoors	363,797	0.6	334,973	92	4.4	156,850	43
4	Indoors	406,092	1.32	215,872	53	5.3	79,371	20

## Data Availability

Data are available from the author on request via lwallace73@gmail.com.

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
