# Peer review of "Intercomparison of PurpleAir Sensor Performance over Three Years Indoors and Outdoors at a Home: Bias, Precision, and Limit of Detection Using an Improved Algorithm for Calculating PM2.5"

_sensors, 2022, doi:10.3390/s22072755_

Round 1

Reviewer 1 Report

The paper entitled " Intercomparison of PurpleAir sensor performance over three years indoors and outdoors at a home: bias, precision, and limit of detection using an improved algorithm for calculating PM2.5" deals with a very important and actual topic. Namely, analysis provided clearly identifies problems of the CF1 and CF_ATM algorithm and states advantages of the CF3 algorithm in the case of low concentration. The materials and method are relevant and very well chosen for the topic analysed. The author very precisely described each aspect of the study and step by step explained advantages of CF3 algorithm, and simultaneously drawback of CF1. The result presented in the paper could be highly beneficiary for every research working in these field and for improving results of future measurements. In my opinion the paper could be accepted for publications in the present form with few small clarifications and comments listed below.

It it not clear how comparision with FEM bias is related to this research. More detailed explanation about is preferable.

It there any variation in the interoperability of the sensors depending on the period of the year or air temperature and humidity if measured, i.e. does any of these parameters influence accuracy of measurements and test performed? 

Could the authors elaborate in the conclusion if the problem with LOD could influence the results of measurements in the indoor and outdoor areas with higher pollution or this affects only lower polluted areas. .

Some related papers like the following few have not been cited, so please add them to the reference list.

Badura, M.; Batog, P.; Drzeniecka-Osiadacz, A.; Modzel, P. Evaluation of Low-Cost Sensors for Ambient PM2.5 Monitoring. J. Sens. 20182018, 16.

Zaric N, Spalevic V, Bulatovic N, Pavlicevic N, Dudic B. Measurement of Air Pollution Parameters in Montenegro Using the Ecomar System. International Journal of Environmental Research and Public Health. 2021; 18(12):6565.

Lelieveld, J.; Klingmüller, K.; Pozzer, A.; Pöschl, U.; Fnais, M.; Daiber, A.; Münzel, T. Cardiovascular disease burden from ambient air pollution in Europe reassessed using novel hazard ratio functions. Eur. Heart J. 201940, 1590–1596. 

Reviewer 2 Report

The authors present a work on collecting and analyzing data from 4 different systems (PurpleAir) mounted in a house for a period of up to 3 years.  While both the data and the comparison of the post-processing algorithms might be useful and of practical value, the work itself could be refined to be closer to a research article.

The authors are encouraged to reshape the manuscript such that it includes an introduction section concerning the general problem, the motivation of the work, the impact of the work. A brief (or extended) review of the state of the art could be fruitful.

The methodology of the work should be better described both for the algorithmic process and the analysis. The mathematical formulation should be used for the description of relevant parameters/indicators.

Verification and validation of the methodology should be included before the result description.

The outcome of the research could be provided both in tabular and illustrative (e.g. Figures) way to support the reader.

A discussions section could be added in case the authors feel like there is an issue to be discussed (e.g. regarding the precision or the bias of the system).

Concluding remarks could include an overview of the work and the main outcome but also a description of the added value the works provide among its innovation.

In general, authors should adopt more formal verbatim.

I believe that the work could not be published in the present form.

Reviewer 3 Report

First of all, I would like to congratulate the author as the challenge of this study mainly stay in the extremely long run of the field experiment. Unfortunately, while I was expecting quite a lot with this title, I was quite disappointed going throw the document. In particular as this paper seems to be a field application of an interesting paper published last year by the same author. Lastly, le last sentence in line 193-194 ("The ALT-CF3 algorithm, available on the PurpleAir API site, is the only one there that does not depend on the Plantower proprietary algorithms."), gives the impression that the author is publishing a manuscript against the Plantower company, a feeling already perceptible in the end of the introduction, whereas the subject of the study is an impressive 3 years real-life intercomparison of the PurpleAir systems.

Comments:

Line 6: Is it normal that you did not mention any institute, laboratory, company…? Usually the author must give information about the institution/laboratory where the work has been carried out. In particular, we can find references from the same author in this manuscript where "Independent Researcher" is mentioned.

Line 17: "The percent of observations", maybe you meant percentage?

line 23-25: "Their behavior in outdoor sites has also been studied extensively [7-12])... However, fewer studies have focused on their use indoors [15-21].", pay attention, you mentioned 5 references for outdoor against 6 references for indoor saying that "fewer studies" have been carried out for indoor uses.

Line 34: "the need to maintain, clean" maybe you meant "the need for maintenance, cleaning"?

Chapter 2 "Materials and Methods": In the introduction, you mentioned precision and accuracy. Those parameters are usually calculated against a reference instrument. What is the reference you are using in this study?

Line 61: "desk or dresser about 1.6 m high", which use did you targeted? in wich room?

Line 61: "The outdoor monitor was hung about 2 m high from a bracket about 15 cm from the home", I would advise to add a picture of the mounting system in order to ease the understanding, or maybe a simple scheme.

Line 66: "The air exchange rate of the home was measured on multiple occasions and was generally in the range from 0.2 to 0.3 air changes per hour", what is the unit of this air exchange rate ? can you provide it in some flow unit ?

Paragraph 3.1 "Relative Bias": in order to ease the understanding, I would adivce to gather table 1 and 2 in one only table to ease a sensor-to-sensor comparison.

Line 104: "FEM or FRM monitors" can you please give the definition of both acronyms before citing in the manuscript?

Line 116-118: "These results suggest that the range of the absolute bias of 33 PurpleAir monitors employed in [32] using the CF3 algorithm appears comparable to the absolute bias of the 61 FEM monitors compared to FRM monitors in the EPA comparability assessments." In order to fully compare both measurement bias, you should first ensure that the pollutant matrix are at least similar (same pollution source, same PM concentration, same PM granulometry, same temperature range, same relative humidity range...) as all those parameters can have an important impact on the performances of the devices

Line 159: "One approach was advanced in", maybe you meant "One approach was proposed in" ? In french the word "avancé" can be used in this context, but its literal translation "advanced" do not have the same meaning of "proposed".

Line 177: "At typical levels" in order to have a better understanding, you should provide in the manuscript a table with mean, max, min and maybe median of the measured levels of PM2.5

Line 184: "Mean precision using the CF3 algorithm" CF3 or ALT-CF3?

Line 185: "much worse at about 7-8%", there is only 1% difference between 7-8% against 6-7%, "much worse" seems really exaggerated in this case. In particular when you write "was not greatly worse" in the following sentence when comparing ~4% to ~3%.

Reviewer 4 Report

The article under review concerns the evaluation of Particulate Matter (PM) sensors in indoor environment. The author used in this study eight PMS5003 by Plantower designed for measuring PM2.5 concentrations. The investigation conducted in this work is aimed to prove that the algorithm used by the sensor manufacturer for calculating PM concentrations  is not appropiate for indoor use, because it assigns zero values to measures which fall below a threshold. To overcome this issue, the author proposes an alternative algorithm (the ALT-CF3) , and he states that through the use of it, the precision of sensors is increased. No reference monitors were used in this study, therefore it is impossible to say if the low values of PM concentration detected by the sensors are really close to zero, or they were assigned to zero even though they were not close to zero. In other words, the absence of reference instrumentation seriously affects the scientific rigorousness of the investigation. The improved precision of the ALT-CF3 algorithm should be proved by comparing data measured through the factory algorithm and data calculated by the new algorithm against data provided by regulatory-grade, or scientific-grade  monitors, or at least against instruments featured by higher accuracy than the devices under test. This consideration gets more importance, especially if the study leads to the strong statement through which it is declared that the device use is not appropriate in indoor environments, and also, if we consider that previous works  (AQ-SPEC: http://www.aqmd.gov/docs/default-source/aq-spec/field-evaluations/purpleair---field-evaluation.pdf) have proved a strong correlation between the sensors and reference instruments (even though in outdoor environments the PM concentration levels are in general not close to zero).  Anyway, I agree with the author on the fact that  the algorithm used by Plantower is quite obscure, and it is not helpful for the device performance assessment. At last, I would like to stress that I'm not questioning the validity of the proposed algorithm, but I strongly suggest the author to prove the increased precision achieved through it, by conducting the experiment in co-location with reference monitors, and by comparing outputs of sensors with data provided by reference using standard indicators such as Root Mean Squared Error, Standard Deviation etc.

Round 2

Reviewer 2 Report

Thanks the author for the updated version. While the refinement of the manuscript is in the right direction, the main points of the previous review are still valid. 

The work would need further major refinement in order to be closer to a research paper rather that a technical report. 

Author Response

The reviewer made no new comments in his 2nd review--only that the paper struck him/her as closer to a technical report than a research paper.  Having written several hundred peer-reviewed papers and scores of technical (government) reports, I have a sense of the difference between them.  Research papers are concentrated short (7-10 journal pages) distillations of research with attention paid to previous work in the field.  Technical reports are typically hundreds of pages long without much reference to previous work since they are meant to provide a complete record of the work done by the submitting corporation or research institute.  

Reviewer 3 Report

Again, I would like to congratulate the author for the work carried out and presented in this paper. The improvement brought to this reviewed version added a lot of interest. Please find below some last comments.

Paragraph 2: thank you for the pictures which help understanding. However, we can see that the door devices have been mounted upside down, i.e. with the sensor inlet facing up whereas for the outdoor device the inlet seems to be facing down. Can this have a possible impact on the measurement performances of the sensors? is there any mounting advice from the manufacturer?

Line 71: "vaping pen containing marijuana liquid" does the nature of the liquid have an impact on the particle distribution or PM mass?

Figure 2: the legend is unfortunately not complete for outdoor

Line 146: "never returs", I guess it is only a typo and you meant "never returns"

Reviewer 4 Report

dear author, the article "Secondhand
exposure from vaping marijuana: Concentrations, emissions, and exposures determined
using both research-grade and low-cost monitors", cited in this study, in which findings are supported by comparison with research-grade monitors is an intersting and scientifically valid work. However, we are assessing another study, in which new algorithms for measurements performed by low-cost sensors in another environment (the indoor one) where different conditions are present. With all the due respect, but in my opinion, any study of this type should be conducted by comparing the algorithm, or the device under test, with reference instrumentation of higher accuracy than the device or algorithm under test. Therefore, I strongly suggest to repeat the experiment (few weeks should be enough) in the modality early mentioned. Otherwise, I'm so sorry, but I don't feel to approve the methodology adopted for this study.

Author Response

I thank the reviewer for his comment regarding use of a research-grade instrument in these studies.  I agree that one is absolutely necessary when attempting to discover the accuracy of a particular instrument or algorithm.  Not only is a research-grade instrument necessary, it must itself have a calibration factor that is suited to the aerosol mixture being sampled.  My colleagues and I hve many publications in which we use gravimetric lab-based filter weighings on microbalances to establish the accuracy of the instrument.  

However, this study does not attempt to establish the accuracy of the PurpleAir monitor.  We stated from the beginning that we do not know the absolute bias of the instrument with respect to indoor air, other than the single example from the study supplied to this reviewer of indoor vaping of marijuana (Wallace et al., 2021).  That study used real-world measurements in an occupied home and found that the aerosol mixture had essentially the same calibration factor as was found for 33 outdoor PurpleAir monitors within half a kilometer of 27 regulatory monitoring sites.  Although the aerosol mixture being studied was influenced mainly by exhaled marijuana, there was also an unavoidable component of the mixture due to infiltration from outdoors.  The expereiments lasted from 6 to 20 hours, so after the initial peak due to marijuana had dissipated somewhat, the later hours of each experiment would have been measuring increasingly larger relative contributions from outdoor air.  Each of the 47 experiments was run alongside two research-grade instruments, the SidePak and Piezobalance, which themselves were backed up by 8 experiments using gravimetric sampling, which established CFs of 0.44 for the SidePak and 0.97 for the Piezobalance.  The experiments took one year to complete, and at the end of that time we had a calibration factor for the indoor aerosol mixture of 3.0., which happened to match nearly exactly the outdoor CF of 3.05+- 0.05 found for the 33 outdoor monitors.  This is a somewhat long-winded way of making the point that it is not enough to have a research-grade instrument along side the instrument being tested unless we have established a calibration factor for the research-grade instrument that is based on the aerosol mixture being sampled. Apart from cigarette smoke (which is not an issue for this study since the home occupants are nonsmokers) there are not a lot of known calibration factors for typical indoor air sources.   The single most important calibration factor to have would be that for outdoor air, since it forms a substantial contributor to indoor air at all times. Fortunately, we have that calibration factor for the ALT-CF3 algorithm.  

But unfortunately, the research-grade instrument available for this study (the SidePak) has been examined for an outdoor CF, and the results in the one major study (Jiang, 2011) covered such a range (0.31 to 1.05) that Jiang et al concluded that outdoor air might be so changeable in refractive index, density, etc. that no single calibration factor might be applicable universally.  So at least for outdoor air, the PurpleAir monitor itself may have the best CF when cmpared to the research-grade instrument.

A few other studies have established a CF for the SidePak for some typical indoor sources.  The major study here is Dacunto (2013), which established SidePak CFs for wood burning in a fireplace, burned toast, hamburger, salmon, etc.

In response to this reviewer, I have completed experiments looking briefly at 18 or 20 common indoor sources and using a co-located SidePak.  (Actually, 3 SidePaks alongside three PA-II monitors (six sensors)).

I have added a brief discussion of the materials and methods and results to the main paper and more has been added to the Supplementary Information. The ALT-CF3 and Plantower algorithms were applied to these results, with similar findings regarding the relative precision of the two algorithms.

Round 3

Reviewer 2 Report

General Comments:

Air quality monitoring approaches based on low-budget systems, like the one discussed in the work, are becoming very popular mainly due to their open data architecture and the dense network. The topic of the work is very interesting and similar efforts are very important for the in-depth understanding of borders and limitations on the issue.

In general, the manuscript has been significantly improved since its first version. The authors managed to make the work more concise and robust. A round of some further refinements could enhance its impact.

 Specific Comments:

  1. The authors could consider refining the lines 10 - 11 of the abstract such that the questions included are more passive. For example, the questions “Are they reliable?...” could be replaced by a phrase like “Despite their impact questions are raised regarding their reliability and their performance, especially for long-term operation…
  2. In addition, reference on specific company and product and question on its reliably could (line 11) be avoided. For example, authors could make the question more generic and then state that in the current work the evaluation of …..
  3. While authors present a brief review of the current state of the art (lines 23-33), they have excluded recent works that are focusing on the evaluation of similar approaches in the real environment. Authors are encouraged to add similar approaches to their manuscripts.

[1] Spandonidis, C., Tsantilas, S., Giannopoulos, F., Giordamlis, C., Zyrichidou, I., & Syropoulou, P. (2020). Design and Development of a New Cost-Effective Internet of Things Sensor Platform for Air Quality Measurements. Journal of Engineering Science & Technology Review, 13(6).

[2] Mahajan, S.; Gabrys, J.; Armitage, J. AirKit: A Citizen-Sensing Toolkit for Monitoring Air Quality. Sensors 2021, 21, 4044. https://doi.org/10.3390/s21124044

  1. The authors could consider adding the purple air site in references following Journal Formatting (e.g. day accessed etc). The same applies to companies referred to in lines 78-79 and 184.
  2. In line 45 the phrase developed nations could be avoided or further described.
  3. Statement in lines 44-48 regarding the unavailability of data from the company side could be refined. It is expected that a company will not resolve specific details on its product, especially in an emerging market that shares an annual portion of almost 2 billion USD.
  4. The paragraph in lines 57-59 could be removed to a place that could suit better. Instead, a paragraph describing the structure of the paper could be added in these lines.
  5. The description of the experimental apparatus (lines 62-72)is very illustrative. Authors could consider adding a relevant diagram and/or some images that could depict the installed sensors in their real environment.
  6. On page 3 some manuscript editing is needed regarding the extended gap.
  7. Authors could consider adding a relevant flowchart in section 3.1
  8. Authors should take extra care when using a document from another work (lines between 263 and 264).
  9. Authors should refine the conclusions such that they describe not only the main finding but also illustrate the novelty of the work as well as its impact. Limitations and borders should also be emphasized (if any).
  10. Authors could consider refining the figures such that they are better suited within the text. For example, Figure 3 is too large while including limited information.
  11. Authors could refine the manuscript such that all abbreviations are described the first time they appear in the text. There is no reason to be described again further down to the manuscript.

Reviewer 4 Report

dear author, I want to thank you for your effort to improve the article by following my suggestions. I'm still convinced that any experiement should be checked through a suitable reference method, system, or instrument. Anyway, considering your modifications to the manuscript and, most of all, your previous works. I agree to publish this article. Congratulation for your hard work.

Author Response

I thank the reviewer for his useful comment regarding comparison with a reference instrument.  I did carry out these comparisons for 18 or so indoor air particle sources and they provided some additional insights.